# Improving the 3D Printability and Mechanical Performance of Biorenewable Soybean Oil-Based Photocurable Resins

**DOI:** 10.3390/polym16070977

**Published:** 2024-04-03

**Authors:** Marius Bodor, Aurora Lasagabáster-Latorre, Goretti Arias-Ferreiro, María Sonia Dopico-García, María-José Abad

**Affiliations:** 1Campus Industrial de Ferrol, Grupo de Polimeros-CITENI, Universidade da Coruña, 15403 Ferrol, Spain; marius.bodor@udc.es (M.B.); goretti.arias@udc.es (G.A.-F.); s.dopico@udc.es (M.S.D.-G.); 2Dpto Química Orgánica I, Facultad de Óptica y Optometría, Universidad Complutense de Madrid, 28037 Madrid, Spain; aurora@quim.ucm.es

**Keywords:** AESO, bioresin, DLP, 3D printing, acrylic resin, reactive diluents

## Abstract

The general requirement of replacing petroleum-derived plastics with renewable resources is particularly challenging for new technologies such as the additive manufacturing of photocurable resins. In this work, the influence of mono- and bifunctional reactive diluents on the printability and performance of resins based on acrylated epoxidized soybean oil (AESO) was explored. Polyethylene glycol di(meth)acrylates of different molecular weights were selected as diluents based on the viscosity and mechanical properties of their binary mixtures with AESO. Ternary mixtures containing 60% AESO, polyethylene glycol diacrylate (PEGDA) and polyethyleneglycol dimethacrylate (PEG200DMA) further improved the mechanical properties, water resistance and printability of the resin. Specifically, the terpolymer AESO/PEG575/PEG200DMA 60/20/20 (wt.%) improved the modulus (16% increase), tensile strength (63% increase) and %deformation at the break (21% increase), with respect to pure AESO. The enhancement of the printability provided by the reactive diluents was proven by Jacobs working curves and the improved accuracy of printed patterns. The proposed formulation, with a biorenewable carbon content of 67%, can be used as the matrix of innovative resins with unrestricted applicability in the electronics and biomedical fields. However, much effort must be done to increase the array of bio-based raw materials.

## 1. Introduction

Nowadays, there is a growing trend to replace petroleum-derived plastics with renewable resources to reduce their environmental impact and promote the circular economy. This is particularly challenging for new technologies such as vat photopolymerization 3D printing due to the limited availability of environmentally friendly photocurable polymers [1]. In recent years, new biomaterials have been modified and tested for photocurable printing, such as polyethylene glycol or polycaprolactone [2,3,4] or biomass compounds such as vegetable oils, polysaccharides, lignin, terpene-derived compounds or proteins [5]. Specifically, vegetable oils (VOs) are frequently used in manufacturing industries because of their abundant C=C bonds which are simple to epoxidize; epoxidized vegetable oils (EVOs) are abundant, economic, renewable, sustainable, nontoxic and environmentally friendly [6]. Bio-based modified vegetable oils have been shown to improve the flexibility of UV curing resins in order to cause a similar effect on photocurable resins for 3D printing [7]. Among vegetables oils, soybean stands out due its high production worldwide (60% of total oilseed production) [8,9]. The epoxidized soybean shows excellent thermal photostability and good inter-solubility with resin materials [10]. The acrylated derivative (AESO) can be cross-linked via UV-induced free radical polymerization, resulting in a highly crosslinked polymer [11]. Its properties vary depending on the number and type of polymerizable groups, methacrylate or acrylate [12]. Lebedevaite et al. studied mixtures of initiator-free AESO-based resins with acrylate plant-derived monomers as photosensitive resins for the direct laser writing lithography technique [13]. However, its printability by digital light processing (DLP) or stereolithography (SLA) can be hindered by its high viscosity. These vat polymerization techniques are based on the layer-by-layer solidification of a liquid photosensitive resin via UV-light exposure. If the viscosity of the resin is too high, the liquid formulation cannot fill the gap between the vat and the previous polymerized layer homogeneously, causing the failure of the printing process [14]. Moreover, lowering its viscosity is essential to add fillers that improve the properties of the final materials or provide them with a specific functionality [15,16,17]. Therefore, several approaches are being explored to reduce its viscosity and enhance the performance of the resin and the cured materials. For example, compounding the AESO through dual-curing hybrid systems based on urethane epoxidized soybean oil (SBO-URE) and epoxy acrylates [7] or the development of a vitrimer based on acrylated soybean oil and a monomer derived from glycerol [18].

Other alternatives are based on the effect of the diluents, both non-reactive and reactive. Non-reactive diluents such as ethanol [19,20] or soybean oil [21] have been used to reduce the viscosity of the resin and even allow the addition of a filler [15,16,17,19,20]. The use of reactive diluents has the advantage of tuning the properties of the material other than reducing the viscosity of the resin and improving its handling. Although AESO can be UV cured without any other comonomers [15,16], the addition of small amounts of a crosslinking agent, such as polyethylene glycol diacrylate or polycaprolactone diacrylate, has shown to facilitate its photopolymerization and control the final properties of the polymer networks, such as tensile strength and elongation at the break [17]. For DLP or SLA applications, usually (meth)acrylates with functionality up to three are employed, such as hydroxyethyl methacrylate (HEMA) [22], isobornyl methacrylate (IBOMA) [12,23], isobornylacrylate (IBOA) [24], tetrahydrofurfuryl methacrylate (THFMA) [12], 1,6-hexanediol diacrylate (HDDA), [10,25] polyethylene diacrylate (PEGDA) [26,27,28], trimethylolpropane triacrylate (TMPTA) [10,25] or other compounds such as acryloylmorpholine (ACMO) [10]. A summary of these references has been included, although comparisons between them are difficult due to the different experimental conditions and products used, including the origin of the soybean acrylate oil, either commercial or specifically synthesized for the study.

Guit et al. [12] synthetized di- or trifunctional soybean oil acrylates and methacrylates from epoxidized soybean oil and mixed them with the bio-based diluents, IBOMA or THFMA. IBOMA was selected as the diluent for both acrylate or methacrylate soybean oil formulations (60 wt.%) for DLP printing, considering the viscosity of the resin and the mechanical performance of the material on the casting probes (storage modulus 727–1007 MPa). Rosa et al. [29] also combined IBOMA with a commercial AESO so that the diluent increased the tensile strength but decreased the elongation at the break; the AESO/IBOMA 50:50 composition led to mechanical properties similar to commercial resins. The IBOMA content further decreased the critical energy necessary to start the polymerization.

Zhu et al. [22] combined a bio-based UV-curable oligomer (GMAESO) with HEMA to obtain a resin suitable for DLP; GMAESO was previously synthesized from epoxidized soybean oil (ESO) and gallic acid (GA). The optimal mixture GMAESO with 50% HEMA showed low viscosities (93 mPa s) and enhanced mechanical properties with a tensile strength of 42.2 MPa.

Barkane et al. [25] compared neat AESO and the ternary mixture of AESO with HDDA and TMPTA with a constant weight ratio of 65/30/5, using 2,4,6-trimethylbenzoyldiphenylphosphine oxide (TPO) as the photoinitiator (PI). The addition of the diluents not only decreased the viscosity of the resins, but also induced a decrease in the photopolymerization time by 25% and an increase in the double bond conversion rate (DBC%) by 10%. Both thermal and mechanical properties were improved because of the increased crosslinking density. Indeed, the storage modulus improved by almost four-fold at room temperature and five-fold at 80 °C with respect to pure AESO, reaching 726 and 72 MPa, respectively, for the formulations containing the reactive diluents. The optimized formulations were printed by SLA and it is noticeably that only those containing the reactive diluents were printable.

Chen et al. [10] also investigated the effect of a dual diluent system on the performance of an acrylate epoxidized soybean oil resin. First, they synthetized AESO by esterifying acrylic acid (AA) with epoxidized soybean oil (ESO). Then, they mixed it with either ACMO/TMPTA or ACMO/HDDA; the best results both for the printability and performance of printed parts were obtained using the dual diluent ACMO/HDDA with a ratio of 3:2 combined with AESO (50% weight). This formulation led to a resin viscosity of 174 mPa and a tensile strength of 52 MPa. The higher functional diluent TMPTA (trifunctional) led to higher viscosity than HDDA (bifunctional). Lastly, the properties of the printed materials were further improved by adding 10% epoxy resin to the aforementioned formulation. For example, the tensile strength increased up to 58 MPa.

In this work we have investigated the effect of several reactive diluents on the performance of commercial AESO for DLP printing, prioritizing bio-based products [30] in the search for the best mechanical performance. Based on previous studies, only mono- or bifunctional reactive diluents were selected to facilitate the reduction of the AESO viscosity. First, the individual effect of the reactive diluents on viscosity and mechanical properties was studied. Then, two dual diluent systems were further investigated considering their effect on the hydrophobic and mechanical properties of the formulation and their printability accuracy. In this way, the resin formulation was modified seeking for the maximum content of bio components, the appropriate viscosity for high-quality printing as well as a range of mechanical properties that allow various applications [17].

The reactive diluents tested were lauryl acrylate (LA) and three di(meth)acrylate derived from polyethylene glycol, namely polyethylene glycol dimethacrylate (molecular Weight = 200) (PEG200DMA) and two polyethylene glycol diacrylates with molecular weights of 575 and 700, respectively (PEG575DA and PEG700DA). Although PEG575DA and PEG700DA were not bio-based, they were included in the study to evaluate the effect of their molecular weight and hydrophilicity on the performance of the material [30]. Further, the use of Food and Drug Administration (FDA)-approved biopolymers, which had been employed in the biomaterials industry for many years, will allow for the expansion of the application of these types of resins in the field of biomedicine [26,31]. Finally, low-molecular-weight PEGs (MW ≤ 7400 Da) have the additional advantage of being completely biodegradable both in freshwater media and sea water [32].

## 2. Experimental Section/Methods

### 2.1. Materials and Samples Preparation

Acrylated epoxidized soybean oil (AESO) (Mw = 1200 g/mol), polyethylene glycol diacrylates with average molecular number (Mn) of 550 and 700 (PEG575DA and PEG700DA, respectively) and the photoinitiator diphenyl (2,4,6- trimethylbenzoyl) phosphine oxide (TPO, molecular weight = 348.37 g/mol) were purchased from Sigma Aldrich (St. Louis, MO, USA). Ethanol and 2-propanol were obtained from Scharlau (Sentmenat, Spain). C12-Alkyl Acrylate or lauryl acrylate (LA) (SARBIO 5101) and Polyethylene glycol dimethacrylate (Mw PEG unit = 200) (PEG200DMA, SARBIO 6201) were kindly donated by Sartomer (Arkema, France). The commercial resin ELEGOO Water Washable Resin, Clear Blue, was purchased from ELEGOO. All chemicals were used without further purification.

**Preparation of the resin formulations.** AESO was mildly heated in a water bath at 50 °C in order to reduce its viscosity and promote the mixing with the diluents and initiator. The diluents were then added at increasing ratios and the blends were further mechanically stirred at 50 °C for 30 min. For each type of diluent, the largest amount added was limited by the solubility in the AESO. Subsequently, the TPO photoinitiator at 1 wt.% of the total resin weight, previously dissolved in the minimum ethanol volume (4–5% *w*/*m*) in an ultrasonic bath, was poured into the mixture and stirred for another minute. The homopolymers of AESO and those of the reactive diluents were also prepared for comparison purposes. The mixtures were stored in the dark until used to prevent premature curing.

The composition, the biorenewable carbon content (BRC%) and the viscosity at 1 s^−1^ of the prepared preliminary formulations are compiled in Table 1. The BRC% has been calculated according to Equation (1) [16]:(1)$$\mathrm{BRC}\;\%=\frac{Biosourced\;Carbon}{Biosourced\;Carbon+Fossil}\times 100$$

**Printing of selected formulations.** The selected formulations were printed (the ones marked with an asterisk in Table 1). The criteria for this selection, discussed in detail in Section 3.1, were viscosity values, ease of printability and mechanical properties. The homopolymers of AESO and RD were also printed as references. Just before their use, the formulations were further mechanically stirred for a couple of minutes in the water bath at 50 °C and then printed with a 3D DLP printer (ELEGOO mars PRO; wavelength 405 nm) with a light intensity of 9 mW.cm^−2^. Flexible dog-bone-shaped specimens with a thickness of 2 mm, according to ISO 527:2014 [33], were prepared to carry out the different tests; the exposure time per layer varied between 4 and 8 s depending on the formulation. Bottom exposure, corresponding to the first 5 layers, is subjected to a longer exposure time (15 s) to ensure that the sample adheres to the metal platform. The layer thickness (z) was set at 100 µm. The detailed chemical formulas of AESO, reactive diluents and the photoinitiator used to prepare the precursor solutions and a diagram of the preparation process for 3D printing are depicted in Figure 1.

After printing, the samples were soaked in 2-propanol for 15 min to remove the non-cured monomers. A post-curing process was performed with a post-curing lamp (Form Cure, Formlabs, Somerville, MA, USA) for 10 min at 35 °C.

A second group of specimens were printed as a complex porous structure in order to compare the printing accuracy and the quality of the surface finish. A square shape with sides of 20 mm and a thickness of 2 mm, containing 1 mm-side square orifices at a 1 mm distance apart, was printed in triplicate for each resin mixture, with a layer thickness of 100 µm. The time for the 5 base layers used in printing varied from 8 s/layer for commercial resin and the ternary resins to 12 s/layer for AESO, while the time for printing the rest of the layers also varied from 1 s/layer for commercial resin, 4 s/layer for ternary mixtures and 8 s/layer for AESO. The printed samples were post-cured in an UV incubator with the temperature set at a constant 35 °C value. The curing time depended on the stickiness of the surface, as this simple test has been used as an estimate of the degree of polymerization [34], an aspect also verified through FTIR analysis. In this respect, the final curing time varied from 15 min for AESO, to 20 min for the commercial resin and 30 min for the ternary mixtures. After printing, the samples were measured and images were acquired using a set-up composed of a light-source and a cell-phone camera with a 3× magnification, with a distance of 8 cm between the sample and the camera.

### 2.2. Sample Characterization

The viscosity of the acrylic composites formulations was determined at room temperature using a controlled strain rheometer (ARES, TA Instruments, New Castle, DE, USA) with a parallel-plate geometry (25 mm diameter, 1 mm gap). The steady shear viscosity (η) was measured in a range of shear rates between 0.3 and 1000 s^−1^.

The Fourier Transformed Infrared (FTIR) spectra from cross-sections of post-cured printed dog-bone samples were recorded on Jasco 4700 spectrometer equipment in the Attenuated Reflectance Mode (ATR) using a MIRacle ZnSe Single-Reflection Horizontal ATR accessory. Five individual spectra were collected for each sample between 4000 and 550 cm^−1^, with a 4 cm^−1^ resolution over 64 scans. The degree of the (meth)acrylate double bonds conversion (DBC%) based on Equation (2) was calculated using the Bruker OPUS^®^ software version 5.5 (Bruker Española S.A, Madrid, Spain). The spectra were subjected to baseline and ATR correction. The band at 810 cm^−1^, assigned to the outside of plane deformation, δ_=CH2_, has been selected for monitoring the evolution of the polymerization. Thus, the degree of double bond conversion (DBC%) was calculated from the decrease in the area of the absorbance band at 810 cm^−1^, normalized to the carbonyl ester stretching band (ν_C=O_) of the acrylic polymer at 1728 cm^−1^, as an internal reference, according to Equation (2) [35]:(2)$$DBC\%=\frac{{\left(A_{810}/A_{1728}\right)}_{t=0}-{\left(A_{810}/A_{1728}\right)}_{t}}{{\left(A_{810}/A_{1728}\right)}_{t=0}}\times 100\%$$

The subscript t = 0 indicates the peak areas of the liquid resin before UV irradiation, while the subscript t indicates the peak areas at time t of irradiation.

Tensile stress–strain mechanical properties were characterized using an Instron 5569 universal testing machine (Instron Canton, Norwood, MA, USA) with a load cell capacity of 1 kN, operating at room temperature and at a cross-head speed of 2 mm/min until failure. At least five dog-bone-shaped specimens were tested to measure the tensile properties according to ISO 527:2014 [33]. The measurement of the hardness of the composites with a Shore D Durometer (Durotech M202, Hampden Test Equipment Limited, Kettering, UK) was carried out on the aforementioned dog-bone-shaped specimens at a distance of ±6 mm from the edge of the material after 15 s of force application. The measurements were taken at 10 measuring points on each sample and the mean values and standard deviations were calculated according to ISO 868:2003 [36].

The water absorption was determined according to ISO 62:2008 [37]. Six specimens were selected for each formulation and were placed in a drying oven at 45 ± 5 °C for 24 h. Then, they were allowed to cool to room temperature in a desiccator before weighing them (±0.1 mg). The dried mass was recorded as W_1_. They were subsequently placed in separate containers filled with distilled water at 23.0 °C ± 1.0 °C. After soaking for 24 h, 48 h, 120 h and 216 h, the specimens were removed from the water, wiping the water off the surface and weighing the mass, and were recorded as W_2_. The formula (Equation (3)) for calculating the water absorption or swelling ratio (*S_W_*) is:(3)$$S_{W}=\frac{\left(W_{2}-W_{1}\right)}{W_{1}}\times 100$$

After 216 h of water immersion, hydrolytic degradation was examined. The specimens were dried in an oven at 50 °C for 48 h to remove the absorbed water. Then the mass was recorded as W_3_ and used to calculate the degradation (*D_W_*) by following Equation (4):(4)$$D_{W}=\frac{\left(W_{1}-W_{3}\right)}{W_{1}}\times 100$$

Surface wetting [28] measurements were carried out with a Theta Lite attention tensiometer (Biolin Scientific, Västra Frölunda, Sweden). The static water contact angle was measured at least five times on dry samples using a 4 µL sessile drop of deionized water as the test fluid at room temperature and the average values are reported. The contact angles (CA) formed by a single drop were recorded by the software program “One Attension” (https://www.nanoscience.com/products/attension-tensiometers/oneattension-software/, accessed on 25 March 2021). Image records were set to 10 s. The angle values formed between the determined baseline and the right and left contact points of the model liquid droplet with the solid surface were measured. The program averages the angle values of the right and left points. The final average contact angle after 10 s of the experimental period was calculated by averaging all of the mean values.

The Jacobs working curve is a method used to determine the behavior of a resin during the DLP 3D printing process by calculating different parameters such as the theoretical penetration depth (*D_p_* in µm) of the UV light through the resin, the theoretical quantity of the energy needed to start the transition of the resin from liquid to gel (*E_c_* in mJ/cm^2^), the actual cure depth of a resin depending on the energy (*C_d_* in µm) and the theoretical quantity of the energy needed to obtain a specific thickness of a polymerized layer from a resin (*E_max_* in mJ/cm^2^). The data necessary to plot a curve for each type of resin are obtained through systematic tests, which are further described. The Jacobs working curves were calculated for AESO and the ternary mixtures to evaluate their printing performance, specifically on the cure depth relative to the curing time and energy used in the printing process. Circular silicon molds with a 3 mm thickness and 13 mm diameter were used. The molds were placed in the printer’s vat, filled with the liquid resin and cured with a constant intensity of 9 mW/cm^2^. The exposure times varied depending on the mixture’s polymerization behavior; the ternary mixtures were in the range of 4–12 s and for AESO, 8–16 s. The unpolymerized mixture was washed away with 2-propanol. To ensure the accuracy of measuring the thickness of the cured sample, the sample was additionally irradiated under a UV lamp (Dymax BlueWave QX4, at 80% power with an intensity of 212 mW/cm^2^) placed 7 cm above for 150 s to finalize the curing. The sample thickness was measured (thickness gauge Millitast 1080, MAHR GmbH, Göttingen, Germany) and corresponded to the cure depth for the specific energy required for curing; every sample was performed in duplicate. Further on, the penetration depth and the energy needed for the starting point of the transition from liquid to solid were calculated from Equation (5).
(5)$$C_{d}=D_{p}\cdot \ln\left(\frac{E_{max}}{E_{c}}\right)$$

The UV-VIS spectra of the monomers employed were recorded on a Jasco V-750 double-beam UV-VIS spectrophotometer (Jasco Analítica S.L., Madrid, Spain) between 300 and 800 nm with a sampling interval of 1 nm and 25 accumulations.

Confocal microscopy was used to determine the surface roughness of the printed composite samples using a PLu 2300 Sensofar^®^ optical imaging profiler. Images were captured using an EPI 10 × N objective, a depth resolution of 2 µm and a lateral resolution of 1 nm. The roughness parameters Ra (roughness average) and Rq (root mean square roughness) were obtained using SensoMaP 5.0.4 Software (ASME B46.1-2019 [38]). At least five measurements were performed for each sample in order to calculate the average values.

For the statistical analysis, data were presented as mean ± standard deviation (SD) and analyzed using Microsoft Excel (Microsoft 365 MSO v. 2401, Redmond, WA, USA). Differences between means were compared by t-student tests and considered statistically significant at *p* < 0.05.

## 3. Results

With the aim of selecting the most suitable diluent, three sets of bio-based polyacrylate copolymers have been formulated by mixing acrylated soybean oil (AESO) with increasing amounts of either a monofunctional or a bifunctional reactive diluent and a constant weight of a photoinitiator. The average bio-based carbon content of the formulations was determined from the BRC% of the individual monomers. The results ranging from 85.4 to 43% are reported in Table 1. Then, a fourth group of dual diluent formulations was further prepared by mixing AESO with two different bifunctional reactive diluents. These diluents were selected based on the viscosity of the pre-polymer formulations and the potential mechanical properties of the printed samples. To evaluate the best-performing formulations, Jacobs’ working curves and printing accuracy were assessed.

### 3.1. Viscosity of the Liquid Formulations

The viscosity of the pre-curing mixture is an important property that should be considered when developing new resins for 3D printing as it can be a limiting factor for printability. The viscosities of well-known commercial fossil-based resins range from 0.2 to 1.6 Pa·s, [39]. In principle, a viscosity below 2 Pa·s is desired to allow the appropriate recoating of the surface of the immersive platform or the last printed layer [12]. Nonetheless, an excessively low viscosity can cause cracking due to shrinkage stress [40].

The viscosity of the commercial AESO used in this study was 10.31 Pa∙s., with Newtonian behavior in the range between 1 s^−1^ and 10^2^ s^−1^. Although lower than previously reported values, this high viscosity hinders the printability of pure AESO making it time-consuming and negatively affecting the resolution of the printed objects [11]. AESO contains acryl and hydroxyl functionalities which participate in hydrogen bonding between the triglyceride molecules and increase the viscosity. In order to reduce its viscosity, LA, PEG575DA and PEG200DMA were tested as reactive diluents. The viscosity values of pure AESO, the reactive diluents and three sets of binary mixtures, AESO/LA, AESO/PEG575DA and AESO/PEG200DMA, as a function of the shear rate at room temperature are depicted in Appendix A, while the viscosities at a shear rate of 1 s^−1^, typical for the printing process, are shown in Table 1. PEG700DA was further tested with the aim of comparing with PEG575DA in dual diluent AESO-based formulations.

LA and PEG200DMA exhibit strong shear thinning behavior, whereas the PEG575DA and PEG700DA, despite having clearly lower viscosity values than AESO, only have shear thinning behavior at shear rates of less than 1 s^−1^ and a Newtonian behavior between 1 and 10^2^ s^−1^.

As expected from the viscosity values of pure monomers, the viscosity of the binary mixtures at 1 s^–1^ (Table 1) exponentially decreases with an increasing amount of reactive diluent. Namely, it diminishes from 0.79 to 0.14 Pa·s, from 1.18 to 0.10 Pa·s and from 1.06 to 0.23 Pa·s, as LA, PEG200DMA and PEG575DA increased from 20 to 60 wt.%, respectively. But, whereas AESO/LA mixtures display shear thinning behaviors from LA contents greater than 30 wt.%., the AESO/PEG200DMA and AESO/PEG575DA formulations only show shear thinning behavior at shear rates of less than 1 s^−1^ and a Newtonian behavior between 1 and 10^2^ s^−1^. Despite this, the observed descents will help to improve AESO processability and allow the renewing of the feedstock material once the platform moves to form a new layer during DLP printing. Reducing the viscosity also prevents air from being trapped in the mixture which is a problem during AESO printing [40].

Considering these results, the maximum amount of AESO was set at 60% for printing AESO/PEG575DA formulations. On the other hand, despite the high BRC% and adequate viscosity values of AESO-PEG200DMA formulations with a 40–60 wt.% methacrylate content, this binary mixture was not selected as the use of a high methacrylate content would increase rigidity too much (as the high modulus and tensile strength of the homopolymer suggested, Table 2) and slow down the polymerization rate, which was not desirable given the printing difficulties of AESO itself [41].

Moreover, being the objective of this work, to achieve the best mechanical performance with the highest possible percentage of bio-based carbon contents, the AESO/PEG575DA resins were reformulated to propose mixtures with three components, namely, AESO, PEG575DA or PEG700DA and PEG200DMA. PEG700DA was also tested replacing PEG575DA in order to evaluate the influence of the molecular weight. The viscosity values as a function of the shear rate of the ternary mixtures AESO/PEG575DA/PEG200DMA and AESO/PEG700DA/PEG200DMA compared with neat AESO and AESO/PEG575DA 40 and 50 are depicted in Figure 1. Due to the diluent effect of PEG200DMA, the viscosity values of both dual diluent formulations are ≅ 20% lower than those of the AESO/PEG575DA40 mixture and they are identical to those of AESO/PEG575DA50, within experimental error. Besides the lower viscosities, the ternary formulations have the additional advantage of a higher BRC% content.

### 3.2. Characterization of Printed Formulations

#### 3.2.1. ATR Analysis

The ATR spectra of AESO-based formulations containing the reactive diluents were analyzed before and after DLP printing in order to confirm the presence of all the components of the mixtures and the final degree of double bond conversion in the as-printed samples. In a similar way, the spectra of each of the individual components were also recorded (Appendix A).

As observed in Figure 2, the FTIR spectra of the AESO monomer (uncured = u) exhibit the peaks associated with the main functional groups; specifically, the stretching vibrations of –OH groups, C=O and asymmetric and symmetric C–O–C bonds in the ester groups are centered at 3442, 1728, 1269 and 1188 cm^–1^, respectively. The bands are also strong at 2923, 2854 and 1450 cm^–1^, which are assigned to the asymmetric stretching, symmetric stretching vibrations and deformations of the C–H bond in the –CH_2_− and –CH_3_− groups, respectively. In addition, the bands associated with the acrylic groups can be identified at the doublet around 1636 cm^−1^ (ν_C=C_, doublet), 1405 cm^−1^ (in plane deformation, scissoring, δ_=CH2)_, 985 cm^−1^ (vinyl groups wagging CH_2_=CH−) and 810 cm^–1^ (=CH out of plane deformation, (δ_=CH_)). All these bands, which are also present in the reactive diluents used, are significantly reduced during UV polymerization, showing that most of the acrylic groups had reacted. The curing process in printed AESO (cured = c) also involves the shift and/or widening of some bands, especially in the fingerprint region and the bands assigned to the asymmetric and symmetric stretching vibrations of the C–O–C group which move 24 and 26 cm^−1^ towards lower wave numbers, respectively. The C–O–C is next to the double bond; thus, the consumption of double bonds has a great effect on the oscillation of the C–O–C bonds [42].

Further, no characteristic bands of the TPO have been detected in any of the formulations, owing to the low concentrations compared to the amount of the monomers, plus the overlapping with the bands of the AESO and RD. It is likewise worth noting that except for the absence of the –OH stretching band between 3600 and 3300 cm^−1^, the main characteristic peaks of the lauryl acrylate (LA) diluent, overlap with those of AESO [25]. As a consequence, the confirmation of the reactive diluent incorporation into the photocross-linked polymer network is difficult in AESO/LA copolymers since the only difference is the lower absorption in the aforementioned spectrum range. As an example, the spectra of the AESO/LA40 formulation before and after polymerization are exhibited in Appendix A.

The spectra of PEG575DA, PEG700DA and PEG200DMA before and after photopolymerization are displayed in Appendix A. Unlike LA, the PEGDAs and PEGDMA have characteristic bands that are not overlapped with the AESO spectrum, above all, the strong bands at 1140 and 1095 cm^−1^ assigned to the asymmetric and symmetric C–O–C stretching vibrations of the ether groups, whose intensities clearly increase with the number of ethylene glycols units. Unlike the C–O– stretching bands of the ester group, those of the ether group do not downshift upon curing. A weak band at 1350 cm^−1^, allotted to –OC–H bending [31], can also be discerned. Furthermore, the relative intensities of the asymmetric and symmetric CH stretching bands change with respect to neat AESO. Upon increasing the number of ethylene glycol units, the intensity of the band at 2854 cm^−1^ increases at the expense of the band at 2923 cm^−1^. Moreover, due to the hygroscopic nature of these polymers, the water bands at 3400 cm^−1^ and 1640 cm^−1^, assigned to the –OH stretching (ν_OH_) and deformation (δ_OH_), respectively, are clearly visible in the polymers spectra. The latter one overlaps with the C=C stretching vibration of the (meth)acrylate groups.

The observation of the bands related to the ethylene glycol moieties allows for detecting the presence of the PEGs’ reactive diluents in the AESO copolymers; it is worth noting that despite the differences in the spectral features between PEG575DA, PEG700DA and PEGDA200DMA before and after polymerization (Appendix A), the spectra of cured AESO/PEG575DA40, AESO/PEG575DA/PEG200DMA and AESO/PEG700DA/PEG200DMA are superimposable, as it is not possible to distinguish between the two diluents in the terpolymers (Figure 2 and Figure 3).

Just like in the spectrum of neat AESO, the almost complete disappearance of all the bands associated with the double bond of the acrylic groups suggested very high degrees of monomer conversion, both in homopolymers, copolymers and terpolymers. Among all these bands, the peak at 810 cm^−1^ has been selected to monitor the photopolymerization process as it does not overlap with any other band of the spectra. The average degree of double bond conversion (DBC%) was obtained for all formulations and reported in Table 2. As expected, high values are confirmed, spanning from 87.8% to 98.8%. These results confirmed that 1 wt.%. TPO is adequate to achieve strong photocrosslinked resins. The lowest degrees of curing corresponded to the homopolymers of AESO and PEG575DA and to AESO/LA copolymers with AESO contents higher than 60 wt.% due to the high viscosity of the formulations that affects chain mobility, slowing down the propagation step of the radical polymerization reaction. In summary, the additions of these reactive diluents have a positive effect on the DBC% of AESO resins. The ATR analysis revealed successful curing, which is an encouraging outcome because the mechanical properties of the printed materials are strongly dependent on the crosslinking density and, thus, on the double bond conversion [25].

#### 3.2.2. Mechanical Properties

To assess the practical application of the 3D-printed AESO-based resins, mechanical uniaxial tensile tests until rupture and Shore D hardness were performed. This is an important issue due to the fact that lower tensile strengths, though comparable to elongation at the break, have been reported for some bioresins compared with current petrochemical polymers [17]. The representative stress–strain curves of each sample are depicted in Appendix A. From the stress–strain curves, the modulus, stress at the break and elongation at the break of the printed AESO copolymers were calculated and are presented, together with Shore D hardness, in Table 2. For comparison purposes, the tensile properties of the corresponding homopolymers were also tested, with the exception of LA due to its great fragility. For the same reason, the Shore D hardness of the LA, PEG575DA and PEG700DA homopolymers was not measured either.

The tensile strength of pure AESO is comparable to some previously reported values that vary in a narrow range between 4 and 5 MPa; on the contrary, elongations at the break data are extremely variable from 20 to 1% [11,17,21,43]. The different origins of the AESO monomer and its viscosity, the variable curing conditions (amount and type of photoinitiator, curing in molds or printing parameters), plus differences in the testing probes and experimental conditions are at the origin of these variations.

As stated in the introduction, the mechanical properties of AESO-based resins can be tuned by the use of different reactive diluents varying from soft to rigid depending on the presence/absence of bulky and rigid lateral groups and on the final crosslinking density [16]. The final crosslinking density depends on the functionality of the comonomers and, within the same formulation, on the DBC [16]. For LA contents between 20 and 50 wt.%, the Young Modulus and the tensile strength of the AESO/LA copolymers exponentially (R^2^ = 0.993) and linearly (R^2^ = 0.96) diminished, respectively. Opposite to this, the elongation at the break is maintained or slightly increased in AESO/LA20 and 30 resins due to the internal plasticizing effect of the C12 flexible chain of LA. In any case, the elongations at the break of the AESO/LA40 and AESO/LA50 samples decrease to almost half the value of the pure AESO polymer, increasing the fragility of these soft rubbery copolymers.

Regarding the Shore D hardness, a linear descent (R^2^ = 0.98) is also detected when the LA content diminishes from 20 to 40 wt.%. The hardness of both AESO/LA40 and 50 is reduced by half compared with pure AESO.

The deterioration of all the tested mechanical properties upon increasing the amount of the monofunctional LA in AESO/LA copolymers is coherent with the plasticizing effect caused by the flexible lateral chain of LA and with the decrease in the crosslinking density as the ratio of AESO acrylic groups to LA acrylic groups descends from 4.44/1 to 0.48/1 [13], leading to a loosely cross-linked polymer matrix. Hence, LA was discharged considering this poor mechanical performance.

Subsequently, AESO was mixed with reactive diluents with two acrylate end groups. The tensile parameters of AESO/PEG575DA40 and 50 do not differ from each other, within experimental error. Further, in coherence with the lower values of PEG575DA, the Young’s moduli and the tensile strengths are reduced by approximately half their values with respect to pure AESO. The values are similar to those reported by Kim et al. for copolymers of AESO with low-molecular-weight PEGDAs [17].

In order to improve the mechanical properties of the bioresins, two dual diluent formulations were assayed, namely AESO/PEG575DA/PEG200DMA and AESO/PEG700DA/PEG200DMA. It is well known that although methacrylates are less reactive than acrylates, their inclusion into photocurable formulations enhances their stiffness and tensile strength [12]. As can be appreciated in Table 2, the partial substitution of PEG575DMA by PEG200DMA not only clearly improves all the parameters obtained from the tensile curves with respect to the formulation with a single diluent and the same AESO content (AESO/PEG575DA40), but also with respect to pure AESO. This was an expected result as PEG200DMA shows a stiffer and stronger behavior than PEG575DA and PEG700DA. The presence of the rigid bulky methyl group in PEG200DMA, together with the slight increase in the crosslinking density, due to the shorter length of the PEG chain with respect to PEG575DA, justify these results. As a result of the increase in the stress and deformation (%) at the break, the area under the stress–strain curve (deformation energy) increases, with this behavior being related to toughness [26].

Further, the tensile performance of the formulation including PEG700DA is poorer than the dual diluent formulation with PEG575DA. The three parameter values calculated from the tensile tests of AESO/PEG700DA/PEG200DMA are equivalent to those of pure AESO, within experimental error. By increasing the molecular weight of the PEG diacrylate diluent, a weaker network is formed, resulting in a decrease in Young`s modulus and tensile strength. Surprisingly, the higher number of flexible ethylene glycol moieties in PEG700DA with respect to PEG575DA, have not contributed to an increase in ductility, contrary to the results reported by Klimaschewski et al. [44].

From another point of view, the very low hardness of PEG575DA accounts for the reduction in this property value by 18% in AESO/PEG575DA 40 and 50 copolymers, compared to pure AESO. Lastly, the Shore D hardness values of the two terpolymers are similar to each other and to that of pure AESO, due to the presence of the dimethacrylate comonomer, and are approximately 1.5 times lower than those of PEG200DMA. The negative effect on the hardness of PEGDAs is counteracted by the increase due to PEG200DMA. Thus, acrylates–methacrylates terpolymers are a good option to improve the mechanical properties of AESO-based bioresins.

#### 3.2.3. Hydrophobic/Hydrophilic Properties

The degree of swelling was investigated as a high swelling ratio may negatively affect the dimensional stability of the printed parts or result in damage, such as a surface break or wrinkles [28]. As displayed in Figure 4, the printed specimens reached the equilibrium with water within 9 days of immersion.

The water sorption depends on the hydrophilicity of the formulation and the crosslinking density [45]. AESO is hydrophobic, although it absorbs ≅ 1% after 9 days of water immersion due to the hydroxyl groups present in its structure [45]. On the other hand, the two PEGDAs and PEG200DMA are hydrophilic, with the water sorption depending on the molecular weight and, therefore, on the number of polar ethylene glycol units [4]. Thus, the addition of 40% and 50% of PEG575DA into AESO increased the water sorption of the printed copolymers ≅ 5.1-fold and 7.4-fold during the same period of time. The greater swelling of the AESO/PEG575DA50 copolymer is also favored by the slight decrease in the crosslinking density because the ratio of AESO acrylic groups to PEG575DA acrylic groups decreased from 0.86/1 to 0.58/1 upon increasing the PEG575DA content by 10 wt.%.

For the same AESO content, the partial substitution of PEG575DA by PEG200DMA, with a lower number of ethylene glycol units and an additional hydrophobic rigid methyl group, directly bonded to the C-C polymer chain, decreasing the water sorption. Thus, the swelling of AESO/PEG575DA/PEG200DMA, with respect to the copolymer AESO/PEG575DA40, decreased by 2% (from 5.7 to 3.7%). Further, the chemical composition is also the most important factor that accounts for the small increase in water absorption by 0.9% after 9 days when replacing PEG575DA with the more hydrophilic PEG700DA in the terpolymer. In addition, approximately 3 wt.% of water absorption indicates neither hydrophobicity nor excessive swelling [28]. Therefore, unlike the AESO/PEG575 40 and 50 copolymers, the dimensional stability of the terpolymers will not be jeopardized by their application in aqueous or biological media, especially when using PEG575DA as the reactive diluent. In addition, all of the tested samples showed mechanical integrity during immersion in water.

The degradation in water was also tested as ester bonds generated during the free radical polymerizations of AESO and PEGs are susceptible to hydrolysis [21,46]. Small weight losses of less than 1% were observed for all formulations after 9 days of insertion in water. However, one-way ANOVA revealed significant differences between the formulations (*p* < 0.05). The AESO/PEG575 50 lost more mass (0.72 ± 0.9) than AESO and AESO 40 (0.57 ± 0.04 and 0.61 ± 0.06, respectively), which in turn lost approximately twice the weight of the two terpolymers, AESO/PEG575DA/PEG200DMA and AESO/PEG575DA/PEG200DMA (0.37 ± 0.07 and 0.36 ± 0.03, respectively). Thus, the addition of PEG200DMA prevented hydrolytic degradation as ≅ 0.3 can be considered negligible. Similar results were obtained by Mondal et al. [28] for an AESO/PEGDA200 50/50 copolymer.

In order to assess the effect of the inclusion of PEGs’ reactive diluents on the wettability of printed samples, static water CA measurements were conducted. The CA of the dual diluent formulations AESO/PEG575/PEG200DMA and AESO/PEG700/PEG200DMA have been compared to those of the homopolymers AESO, PEG575DA, PEG700DA and PEG200DMA and are displayed in Figure 5. In coherence with previous studies, AESO-printed parts have a very hydrophobic surface, as reflected by the high CA (99 ± 5°). This is an expected result because despite having hydroxyl groups, AESO contains long non-polar fatty acid chains that explain the hydrophobicity of its surface [29]. Nevertheless, other authors have found more hydrophilic surfaces in AESO-printed samples, depending on the printing speed and energy that influence the final roughness; smoother surfaces led to lower contact angles and an apparent increase in wettability [43].

The contact angles of the terpolymers are also very high and there are no statistical differences between them or with the CA from AESO, showing high hydrophobicity. This is an unforeseen outcome, owing to the more polar nature of PEGs (72 ± 3, 50 ± 2 and 55 ± 2 for PEG575DA, PEG700DA and PEG200DMA, respectively) and to the fact that the presence of PEGs on the surface is comparable to that inside the samples, as has been proved by ATR-FTIR analysis (results not shown).

### 3.3. Printing Performance of AESO: Polyethylene Glycol Composites

The Jacobs working curves were calculated for AESO and the ternary mixtures to evaluate their printing performance and the effect of the reactive diluent on the printability of AESO. A set of three curves was plotted (Figure 6), representing the logarithmic energy values used and the corresponding cured depth values. The linear regression fit, the calculated equation and the corresponding correlation coefficient (R^2^) were all added into the graph for each set of results. The Jacobs working curves [47,48] offer an insight into the time needed for each mixture to reach a specific layer thickness while printing; a good adhesion between layers is ensured if the cure depth is higher than the layer thickness. Furthermore, the results of this test are also important for understanding the degree of UV light penetration, specific for each resin mixture, thus offering insight into the theoretical possibilities regarding 3D printing in various conditions related to time exposure and/or the energy used in the process.

The reactive diluents clearly showed an increase in the speed of the polymerization reaction. Therefore, to obtain an approximately 200 µm polymerized layer, twice as much time is required when using AESO (8 s for a 180 µm layer and an energy of 65 mJ/cm^2^) than when using any of the ternary mixtures (4 s for ~225 µm with 27 mJ/cm^2^). These differences are even greater when exposure times increase.

As expected, both the penetration depth and the *E_c_* are higher for the monomer AESO (*D_p_* 1318 µm and *E_c_* ~77 mJ/cm^2^) than for the ternary mixtures, more than two-fold for *D_p_* and almost three-fold for *E_c_*. In this way, the terpolymers presented further similarities in both the penetration depth, with *D_p_* in the range of 523–571 µm, and *E_c,_* between ~29 and ~27 mJ/cm^2^, as can be seen in Table 3 [29].

The comparison of these results with the references is difficult due to the differences between the sources of chemicals, experimental settings, including the type and concentration of the photoinitiator, printer design and even the type of results presented. Therefore, for AESO, the *D_p_* obtained in this study is close to the one reported by Rosa et al. [29]. However, the reported *E_c_* was much higher (144 mJ/cm^2^) compared to our study (77 mJ/cm^2^), which may be explained by the differences in the photoinitiator concentration and sample preparation. The effect of the polyethylene glycol(meth)acrylates to speed up polymerization has also been reported for other diluents such as IBOMA [49]. Increasing the IBOMA content from 40 to 60% in AESO mixtures clearly decreased the required time for curing. Then, ~200 µm-thick polymerized layers for 50 mJ/cm^2^ of energy were obtained for 60% IBOMA [49], although these results are not directly comparable to ours due the different photoinitiator used. The slower polymerization of AESO could be explained by the longer chains of the AESO monomer, compared to the reactive diluents, that leads to a higher viscosity [50,51] in addition to its higher UV absorption at the printing wavelength (Appendix A).

In coherence with previous reports [29,52], the obtained higher penetration depth of AESO could be a drawback for accurate printing due to more scattered light, which could result in a lower surface resolution for the printed sample. In this way, the reactive diluents can tune the printability of the resin without using the common dyes usually employed to prevent this effect at the cost of the transparency [53]. It is important to highlight the good printing parameters obtained for the ternary mixtures, despite the presence of 20% of a dimethacrylate, since methacrylates have lower curing rates than acrylates [7].

To assess the 3D printability of the resin, a specific pattern was selected. The structure shape used in assessing the 3D printability of a resin can vary according to the utilization purpose and the capabilities of the resin used in the printing process. For high precision resins, structures with details that can only be evaluated using an electronic microscope are presented in various research studies, [13,54,55], while for resin mixtures with a low precision output, like AESO-based resins, higher dimension structures were used [18,24], which can be easily evaluated with small or without magnification.

In the present work, the printability degree of selected resin mixtures was performed by visually comparing the same 3D-printed shape and measuring the preset dimensions. For a better comparison, a Commercial Resin (ELEGOO Water Washable Resin, Clear Blue) was also used as a reference. The conditions necessary for 3D printing were set according to the manufacturing specifications for the commercial resin and based on the behavior during the curing depth tests for AESO and the ternary mixtures. Considering these results, a pattern was printed with these three resins, AESO and the ternary mixtures to visually evaluate the final printing performance of the material. The printing layer was set to 100 µm (z).

According to the results of the printability tests (Figure 7), besides the increased speed, the precision seems to be also a plus for using the ternary mixtures compared to only AESO. In Figure 7, four samples were selected, representing the best performance for each mixture and a sample printed using a commercial resin. Although all samples present the 1 mm orifices from the pattern, the AESO sample presents the lowest fidelity to the original model and most of the imperfections. Although the same time was used for both ternary mixtures, the one containing PEG700DA seems to be the one with fewer imperfections and the one that is visually the closest to the sample printed with commercial resin.

Regarding the thickness of the printed samples (theoretical thickness 2000 µm), no important differences were observed among AESO-based samples, which were all lower than the sample printed with commercial resin. Therefore, the printing performance of the terpolymers is competitive with commercial resin.

As a final point, the surface morphology of the samples printed with pure AESO and ternary mixtures of AESO and polyethyleneglycol was compared by confocal microscopy considering the Ra and Rq values. In general, all the samples had relatively smooth surfaces, although the roughness of the ternary mixtures was up by about two times compared to AESO (Figure 8). This difference, which is statistically significant (*p* < 0.05), could be explained by the higher heterogeneity of the system caused by the reactive diluents. Copolymers have previously been shown to increase roughness in relation to homopolymers [56]. Both AESO materials and ternary mixtures showed a good correlation between the theoretical 100 µm of the printing layer and the experimental values, as can be observed in Figure 8, indicating that the printed parts had a high resolution. In this way, the dual combination of PEGs has proven to be a good approach to enhance the printability of AESO resins.

## 4. Conclusions

The current study demonstrates that the use of biocompatible polyethylene glycol di(meth)acrylates as reactive diluents is an effective strategy for improving the DLP printability of AESO-based bio-resins by counteracting the two limiting factors that hinder its photopolymerization, namely, viscosity and UV absorbance at the printing wavelength. Specifically, setting the maximum amount of AESO to 60% prevents viscosity problems. Moreover, the copolymerization of the polyethylene glycols acrylates–methacrylates of different molecular weights in the suitable proportion can lead to a wide range of materials with tuneable properties that can expand their applicability in material engineering.

Among all the formulations tested, the dual diluent compositions, AESO/PEG575DA/PEG200DMA and AESO/PEG700DA/PEG200DMA, with ≅67% of a biorenewable carbon content (BRC%), performed best from the point of view of hardness, tensile and water resistance properties. The results revealed that adding PEG200DMA to a mixture of AESO/PEGDA with higher molecular weights increased the mechanical stress without sacrificing elongation. These formulations speed up the printing process and enhance the accuracy of the printed parts, as confirmed by the Jacobs working curves’ parameters and the high resolution of the complex patterns. Further, whereas the tensile parameters of the PEG575DA-based terpolymer were slightly better than those obtained for the formulations with PEG700DA, the somewhat greater accuracy of the printed parts was achieved with the AESO/PEG700DA/PEG200DMA.

Taken all together, the developed resins are a good option as a renewable material for DLP printing, although further research must be conducted to increase the “bio” content and optimize the overall properties to achieve values comparable to fossil-based formulations.

## Data Availability

Data are contained within the article.

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
