# Peer review of "Improving the 3D Printability and Mechanical Performance of Biorenewable Soybean Oil-Based Photocurable Resins"

_polymers, 2024, doi:10.3390/polym16070977_

Round 1
Reviewer 1 Report
Comments and Suggestions for Authors
In this work, the authors investigated the acrylated epoxidized soybean oil (AESO)-based resin formulated with lauryl acrylate, polyethylene glycol diacrylate (PEGDA) and dimethacrylate (PEGDMA) in terms of various properties. The terpolymer formulation own good mechanical performance could be further used as a biobased material by 3D printing. However, there are some issues that need to be addressed before accepting.
1. Many of the data are described without images or figures, which makes it hard to read.
2. The basic mechanism of vat polymerization 3D printing techniques needs to be mentioned in the Introduction, which will then help the reader understand why properties (like viscosity) were studied.
3. The review of the literature could be improved for better organization. And The author needs to show how these literatures are connected to the current work. The last paragraph in the Introduction needs to be improved to illustrate why and what work was done in the manuscript.
4. In Line 180, the author mentioned an estimation of the time difference of different resin systems. The method needs to be shown and illustrate why printed samples from different printing durations could be compared to each other.
5. The supporting information file was not found.
6. There is an ASE0 in Figure 1. Should be ASEO?
7. In Line 331, “The ATR spectra of AESO-based formulations containing the reactive diluents were 331 analyzed before and after DLP printing”. Should it be compared between the formulation and the as-printed samples?
8. In Table 2, As more LA was added, DBC increased but saw a decrease in mechanical performance which doesn't make sense. Any comments on this?
9. In Figure 7, The details of the printed samples are not clear. All the samples showed curved sides, is that because of the camera or the lens, or it was printed like this?
Comments on the Quality of English LanguageEnglish needs to be improved. For examples, Line169, "during" should be "for".
Reviewer 2 Report
Comments and Suggestions for Authors
The comments are attached.

Reviewer 3 Report
Comments and Suggestions for Authors
After review the manuscript, this needs to be improved in specific issues that are detailed following:
- For better understanding of process I recommend to include a chemical scheme for each materials and the process that happens.
- How was determinated the viscosity reported in table 1.
- Which was the criteria for selection of formulations used for printing? (line 160)
- What is it means the ternary resins in lines 177?? Which are the commercial resins? and why the curing time vary according to resin used?
- Please explain deeply the Jacobs working curves and what is it represents.
- Please correct the text in line 295.
- Section 3.2.1 caption must be FTIR analysis, not ATR.
- In line 339 write "asymetric and symetric stretching, but for who functional group corresponds this?
- For asseveration that some peaks are overlapped, is there some references for it? (line 355)
- I recommend to plot the IR spectra in transmittance mode, and plot the whole range of wavelength, i mean do not short the figure between 1800 and 2500 cm-1, due it would be interesting if some groups appears in that region.
- Most of the figures captions are too long, must be indicative of information reported, not describe the information in it.
- Line 419 please correct the text.
- For report references i recommend to follow the Journal Instructions for authors.
- In manuscript authors make reference of figures in Supplementary file, however this file was not attached, so it is needed to include it.
Round 2
Reviewer 1 Report
Comments and Suggestions for Authors
The manuscript has been well improved by the revision, and I have no more comments or suggestions for being accepted. But it seems there is a typo in Line 137, a reference was added before title "2. Materials and Methods" from the file I got?
Reviewer 3 Report
Comments and Suggestions for Authors
Reviewer wish to thanks to authors for consider the most of the recommendations/corrections to previous version, however I still recommend to change the way how the FTIR spectra are plotted, based that in Spectroscopy techniques, for quantification of species or functional groups, the absorbance way is the recommended based on the Labero-Beer law, and when identification of functional groups is needed, the Transmítanse way is the recommended.
